# Transfer learning in DeepLC improves LC retention time prediction across substantially different modifications and setups

Robbin Bouwmeester [1,2,6], Alireza Nameni [1,2,6], Arthur Declercq[1,2], Robbe Devreese [1,2], Kevin Velghe [1,2], Vladimir Gorshkov [3], Pelayo A. Penanes [3], Frank Kjeldsen[3], Magali Rompais[4,5], Christine Carapito [4,5], Ralf Gabriels [1,2] & Lennart Martens [1,2,4,5] ✉

While LC retention time prediction of peptides and their modifications has proven useful, widespread adoption and optimal performance are hindered by variations in experimental parameters. These variations can render retention time prediction models inaccurate and dramatically reduce the value of predictions for identification, validation, and DIA spectral library generation. To date, mitigation of these issues has been attempted through calibration or by training bespoke models for specific experimental setups, with only partial success. We here demonstrate that transfer learning can successfully overcome these limitations by leveraging pre-trained model parameters. Remarkably, this approach can even fit highly performant models to substantially different peptide modifications and LC conditions than those on which the model was originally trained. This impressive adaptability of transfer learning makes it a highly robust solution for accurate peptide retention time prediction across a very wide variety of imaginable proteomics workflows.

Liquid chromatography (LC) is indispensable in mass spectrometry (MS) analysis of bottom-up proteomics[1]. Through the separation of peptides based on their physicochemical properties, LC reduces the complexity of samples presented to the MS instrument, leading to less ionization competition, improved sensitivity for both data-dependent and data-independent acquisitions, and reduced chimericity in fragmentation spectra (MS2)[2,3]. Additionally, retention time measurements provide a crucial additional information dimension for the downstream analysis of acquired data[4,5].

However, leveraging retention time information in downstream analysis requires the retention time of a peptide to be known, either

experimentally from previous observations, or predicted by a model[6,7]. As a result, many such prediction models currently exist, trained with classical machine learning (ML) and deep learning (DL) algorithms[8–13]. These models learn the mapping between peptide sequences (or features derived from these sequences) and empirical LC retention times. Once trained, these models can also generate predictions for peptides not previously observed experimentally. Retention time prediction models have been successfully applied to various tasks in proteomics workflows, such as improving identification confidence[4,5,14], designing more efficient experiments[15,16], identifying chimeric fragmentation spectra[17], and

[1]VIB-UGent Center for Medical Biotechnology, VIB, Ghent, Belgium. [2]Department of Biomolecular Medicine, Ghent University, Ghent, Belgium. [3]Department of Biochemistry and Molecular Biology, University of Southern Denmark, Odense, Denmark. [4]BioOrganic Mass Spectrometry Laboratory (LSMBO), IPHC UMR 7178, University of Strasbourg, CNRS, Strasbourg, France. [5]Infrastructure Nationale de Protéomique ProFI—UAR 2048, Strasbourg, France. [6]These authors contributed equally: Robbin Bouwmeester, Alireza Nameni. ✉e-mail: lennart.martens@ugent.be

creating spectral libraries for data-independent acquisition (DIA) searches[18–21].

Despite their success, the accuracy of peptide retention time predictions is heavily influenced by experimental parameters, including sample pH, stationary phase of the column, pressure, and temperature[22–24]. Even slight variations in these parameters can alter the retention mechanism, affecting elution times and the order in which peptides elute. Consequently, a model trained on a specific LC setup cannot be applied directly to different setups without proper calibration. This limitation is particularly problematic when predictions need to be applied across varied LC setups without the ability to retrain the model entirely for each setup. Moreover, current models can only predict retention times for peptides carrying modifications that closely resemble those in the training data[9,12,13]. For modifications with drastically different physicochemical properties, current models need to be trained on these specific peptide modifications, requiring large amounts of relevant data.

To apply trained models across different setups, a calibration step using indexed retention time (iRT) peptides or other internally identified peptides in the same run is therefore often used instead of model retraining. This calibration assumes that the predicted elution order of peptides is conserved[23,24]. However, even slightly different conditions can change the elution order, thus leading to poor performance with calibration[25].

However, transfer learning (also called model fine-tuning) offers a generic solution to maintain high performance across diverse data sets[26], by initializing a setup specific model with parameters from a previously trained model rather than at random. This strategy is commonly used to adapt models trained on large data sets to smaller data sets with related prediction tasks[27]. While transfer learning has been applied to retention time prediction previously[10,21,28], we here demonstrate in-depth that transfer learning with DeepLC can improve upon both calibration and retraining in nearly every situation, thus proving the general applicability of the approach. Moreover, and importantly, we show that transfer learning allows DeepLC to adapt to newly presented, complex peptide modifications as well as to substantially different LC setups, showing the promise of transfer learning for accurate predictions in even the most challenging applications.

## Results

The "results" section is divided into five evaluations. The first evaluation consists of evaluating the performance of DeepLC with calibration, training a new model, and with transfer learning. The second performs a large-scale evaluation of transfer learning on 474 data sets. The third evaluates transfer learning's ability to adapt to substantially different peptide modifications. The fourth evaluates the ability of transfer learning to adapt to basic LC conditions when initially trained on acidic LC conditions. The last evaluates the generalization ability of transfer learning by predicting retention times of peptides with modifications that are unseen during training.

### Performance evaluation transfer learning

In the first evaluation, four different data sets are highlighted. For each data set, a DeepLC model was fitted using three methods: calibration (mapping predictions to observations), randomly initialized parameters (new model), and parameters learned from a different data set (transfer learning).

To illustrate the performance of these methods, we used the data from PXD030003 (Fig. 1a–c), a medium-sized data set with 27,954 training peptides. This size means there are enough examples to train a sufficiently accurate model from scratch. Indeed, the model starting from randomly initialized model parameters performs better than calibration in terms of its MAE, 95th error percentile, and Pearson correlation between predictions and observations (Fig. 1a, b). This performance difference is expected as model parameters are not fitted

to the data for calibration, and calibration is only expected to perform very well when the assumptions of a perfect conservation of elution order are met. This latter condition, however, is not likely to be met when experimental conditions change as, even under slightly different LC conditions, peptides that are close in retention time can switch places in their retention order.

When the initial model parameters were set using previously fitted values through transfer learning, the model performed better across all metrics compared to the model starting with random parameters (Fig. 1b, c). This improvement is likely due to the previously obtained model parameters being close to optimal. Indeed, the applicability of these previously fitted parameters is also illustrated by the reasonably accurate predictions with calibration, showing that these existing parameters only need slight fine-tuning to the new LC setup.

Even though transfer learning showed top performance in this example, there is a clear correlation between training data set size and the performance of the three methods for prediction, as shown by the learning curves for three different data sets (Fig. 1d–f). Models with random starting parameters underperformed compared to transfer learning, but ultimately achieve better performance than calibration when sufficient training examples are available (here more than 25,000). For a very limited number of training peptides (less than 100), calibration performed the same or better than transfer learning. However, calibration then quickly reached a plateau, while transfer learning continued to improve with more data, easily outperforming both calibration and the models with random starting parameters.

### Comprehensive analysis of 474 data sets

We have so far examined only four data sets to highlight differences between the three methods for prediction. Here, evaluation includes a much larger set of 362 different PRIDE projects and 474 evidence files that were previously analyzed with MaxQuant. This comprehensive set represents a wide range of different proteomics experiments from various labs, with substantial differences in training set sizes.

Observations from the highlighted data sets in Fig. 1 were consistent with the larger analysis (Fig. 2). In all data sets except one, where performance was almost equal, transfer learning outperformed models with randomly initialized parameters (Fig. 2a). As expected, the number of training peptides correlates highly with the observed difference in performance between the two methods for prediction. For example, data sets with 1000 to 10,000 training peptides showed a 1–2.5% decrease in RMAE with transfer learning compared to training a new model, while larger data sets with more than 10,000 peptides showed a more modest improvement of between 0% and 1% decrease in RMAE.

Transfer learning also outperformed calibration in most cases, with an average RMAE decrease of 1.4%. Data sets where calibration was performed better typically had fewer than 1000 training peptides. Further analysis binned data sets by size, showing that transfer learning and calibration performed equally well for the smallest bin (0–1000 peptides). However, for larger bins, transfer learning showed substantial improvements in RMAE (Fig. 2c). For completeness all data sets with their observed and predicted retention times are separately plotted in Supplementary Figs. 2–18 and performance metrics for data sets are available in Supplementary Data 1.

### TMPP evaluation transfer learning

Although the evaluation on a large number of public domain data sets was comprehensive, it did not test the ability to apply transfer learning in substantially different contexts with minimal training data. This section, therefore, evaluates transfer learning for peptides carrying a TMPP modification (Fig. 3a–c) with a 10-fold CV approach. Due to the size and added hydrophobicity of this modification (572.18 Da), there is a significant change in the retention time of modified peptides.

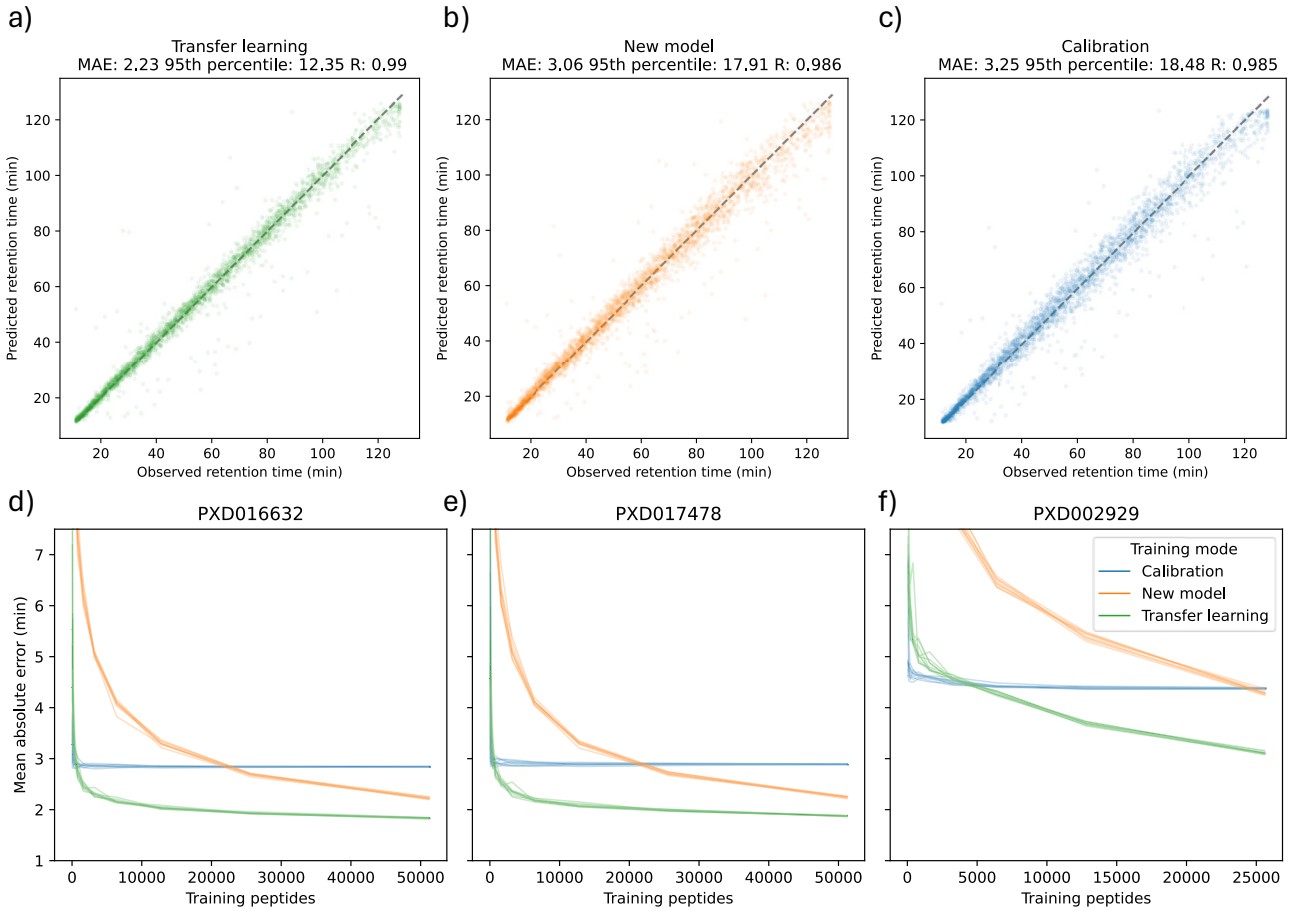

**Fig. 1 | Performance comparison for a selection of datasets.** Where performance is compared for **a** calibration, **b** new model (random parameter initialization), and **c** transfer learning on data set PXD030003. Learning curves for data sets **d** PXD016632, **e** PXD017478, and **f** PXD002929 with 10-fold CV. The y-axis presents the mean absolute error (MAE) and is limited to an upper bound of 7.5.

As expected, retention times predicted with calibration differed significantly from observed retention times for TMPP-modified peptides (Fig. 3a). Over 96% of data sets in Fig. 2 with calibration showed higher correlation compared to the TMPP data set with calibration ($R = 0.82$).

On the other hand, a newly trained model is already able to fit the context of TMPP modifications quite effectively, as the Pearson correlation falls within the best 28% of the data sets presented in the comprehensive analysis presented in Fig. 2. However, transfer learning still outperformed random parameter initialization. The extent of this improvement depended on the number of training peptides, with the largest differences observed for training sets below 2000 peptides (Fig. 3d), but with the MAE continuing to improve for larger training set sizes. The MAE of 0.913 min achieved for transfer learning at 500 peptides was only matched by the newly trained model at 1500–2000 peptides. While there is slight variation in performance when selecting different subsets for training, the order between the three different methods remains mostly consistent across different training set sizes.

### Basic LC evaluation transfer learning
In the previous analysis, the TMPP modification could still be accounted for by a newly trained model, as the underlying retention mechanism is still the same while the model must extrapolate to a novel modification; something DeepLC has been designed to do. In this section, however, the analysis is now performed with a 10-fold CV approach on an LC setup under basic pH conditions. As a result of this drastic change in pH, the retention mechanism changed. Models fitted for acidic LC conditions, which are all models trained up till now, are therefore

unable to predict accurate retention times on this setup. Calibration underperformed markedly in this context, with a Pearson correlation ($R = 0.601$) worse than 99% of data sets analyzed (Fig. 4a). Calibration performs poorly because it is unable to optimize its model parameters to resemble the retention mechanism under basic pH conditions.

Even though the model with calibration is unable to accurately predict for this different retention mechanism, transfer learning is still able to utilize previously fitted parameters. This ability is shown by the improvement of transfer learning over fitting a model from scratch (Fig. 4b, c), which demonstrates that even for substantial differences in data sets, peptide modifications, and LC conditions, fine-tuning a previously fitted model optimally improves performance. Again, as previously observed, the number of training peptides had a significant effect on performance (Fig. 4d). Here, the largest improvements are seen up to around 750 peptides, and a newly trained model only reached similar performance with approximately three times as many training peptides (around 2500). While there is slight variation in performance for small training set sizes, this variation mostly disappears when more than 500 training peptides are used for training.

### Modified peptides evaluation
Previous evaluations demonstrate that transfer learning can outperform both calibration and training a model from scratch. This section evaluates the performance of transfer learning when presented with peptides containing modifications not seen during training (Fig. 5). This data set contains a total of 5693 peptides containing modifications. The count of peptides carrying each specific PTM is indicated below the corresponding PTM in Fig. 5. By subtracting the count of

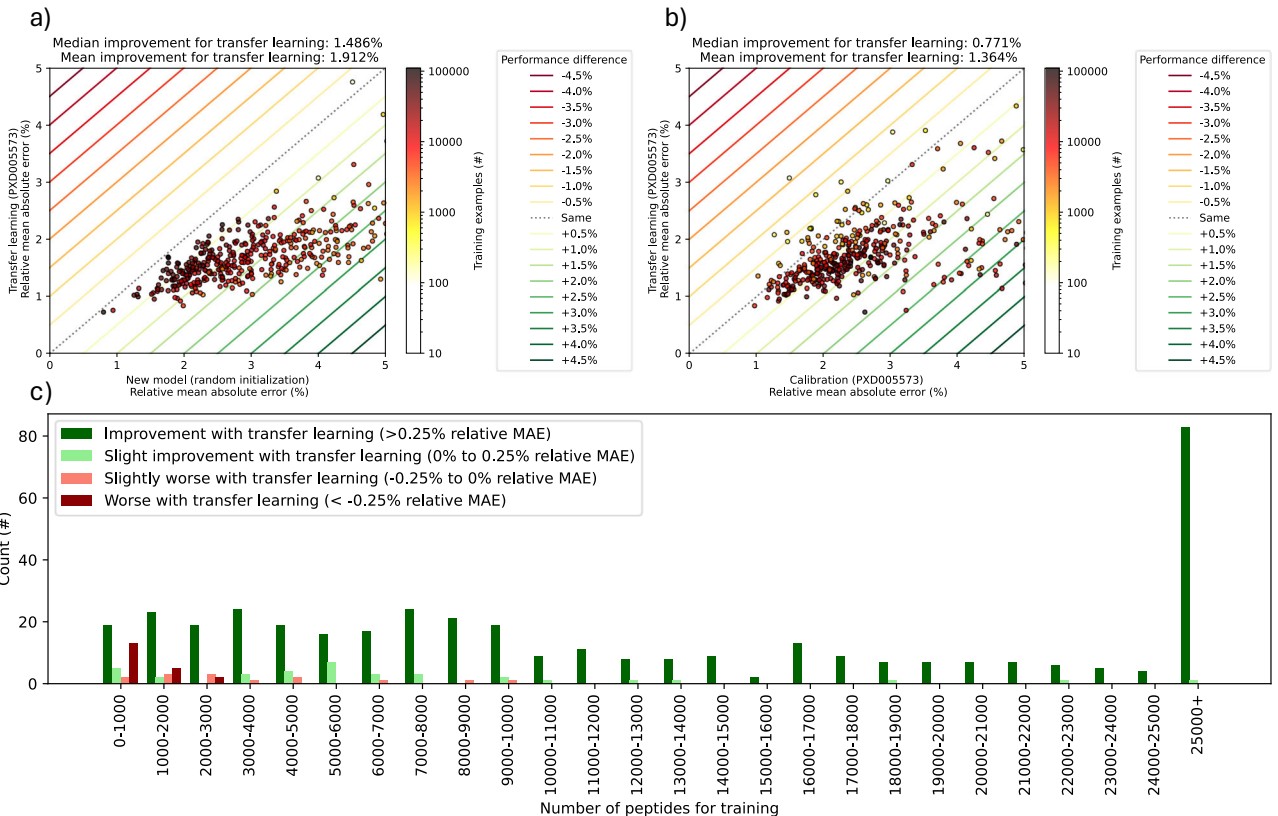

**Fig. 2 | Comprehensive analysis of calibration, new model, and transfer learning performance across 474 data sets. a** RMAE for transfer learning versus a new model (random parameter initialization). **b** Median and mean improvement in RMAE for transfer learning versus calibration. **c** Histogram with data set count for different performance categories (improvement or worsening) for transfer learning compared to calibration, binned by the number of training peptides.

peptides for each PTM from the total number of peptides in the data set, the number of peptides used in training is calculated.

The scatter plot in Fig. 5 compares the mean absolute error (MAE) for transfer learning against a model trained from scratch. There is a modest, but overall, mostly consistent improvement in performance for transfer learning in thirteen out of the fourteen modifications. The only exception is trimethyl where the model trained from scratch shows a much lower MAE. A boxplot showing the residuals confirms these findings between transfer learning and the new model (Supplementary Fig. 19). Furthermore, the results show that calibration has the highest error for twelve out of fourteen modifications.

## Discussion

This study demonstrates the substantial advantages of transfer learning for accurate peptide retention time predictions in MS analysis of bottom-up proteomics. Calibration is unable to maintain accuracy across varied experimental LC conditions, while training new models from scratch requires large amounts of training data. Transfer learning, in contrast, nearly consistently outperforms these methods, showing substantial improvements in MAE, 95th error percentile, and Pearson correlation across a wide range of data sets. The transfer learning approach achieves this by effectively leveraging pre-trained model parameters, thus requiring only fine-tuning to adapt to new contexts, ultimately resulting in highly accurate and robust performance even when provided with only limited new data to train on.

The evaluation of transfer learning across different peptide modifications and LC setups further highlighted its effectiveness in dealing with substantially different chromatographic contexts, showing significant improvements over calibration and new model training. This illustrates how transfer learning effectively generalizes patterns

from previous data to new and diverse modifications and setups. This adaptability makes transfer learning a highly compelling strategy for peptide retention time prediction, offering a reliable and efficient solution to handle the complexities of diverse proteomics experiments. Furthermore, the ability of transfer learning to adapt to new chromatographic setups brings exciting opportunities for experimentalists to introduce more variety in terms of their experimental parameters and conditions. Moreover, we have demonstrated that transfer learning only requires a small number of training peptides to accurately model a new modification's retention behavior. This minimal requirement, for instance, enables the generation of predicted spectral libraries for rare modifications, which can in turn be utilized to explore peptide modifications more thoroughly in DIA data sets. Indeed, even in areas where researchers might have previously felt restricted regarding the applicability of LC prediction models, we have here shown that such limitations are resolved by transfer learning, opening up novel applications for these important tools, and bringing even more flexibility to proteomics experiments.

## Methods
### Architecture
This study uses the same model architecture as was previously presented for DeepLC: a convolutional deep learning framework with four distinct paths for a given encoded peptide. The first two paths consist of convolutional layers that process the peptide's atomic composition (C, H, N, O, P, and S). The third path also uses convolutional layers but relies on a conventional one-hot encoding of amino acids. Finally, the fourth path utilizes densely connected layers to encode global structural features, such as peptide length and the total sum of atoms. All paths are then flattened and passed through six densely connected

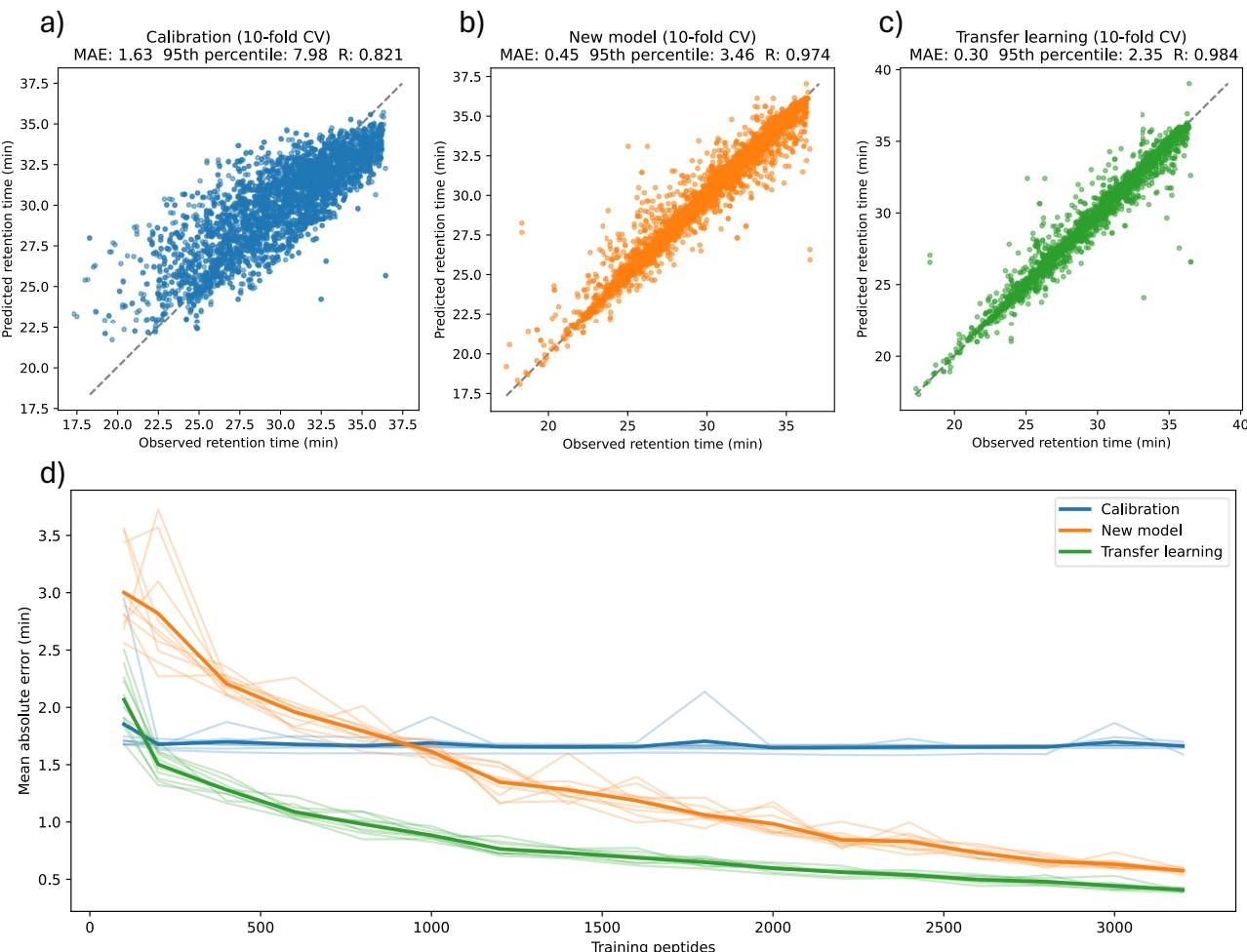

**Fig. 3 | Performance comparison for TMPP-modified peptides.** Evaluation consists of **a** calibration, **b** new model (random parameter initialization), and **c** transfer learning on TMPP labeled peptides. **d** Learning curve showing test set MAE against number of training peptides for TMPP-modified data set. Thin lines represent random seed replicas and thick lines represent the mean of these seed replicas.

layers. A comprehensive description and evaluation of this architecture can be found in the original DeepLC publication[9].

## Data sets and evaluation

To assess the performance of transfer learning with DeepLC, we analyzed 474 peptide identification files from 362 PRIDE projects across a wide variety of organisms (Supplementary Data 1). These data sets were selected because they were previously analyzed with MaxQuant[29] and the resulting evidence files could be downloaded and parsed in a uniform manner. Moreover, many projects were historically analyzed with MaxQuant and it contains information about the elution apex, which is preferred over the retention time of the identification as it is more reproducible. To ensure consistency in retention times across different runs within a single evidence file, an alignment procedure was necessary. This alignment method, detailed on ProteomicsML[30], uses overlapping peptides between runs for calibration. This procedure was performed separately for each evidence file, resulting in 474 internally aligned data sets for downstream usage.

The alignment procedure begins with the run that has the most overlapping peptide sequences with all other data sets. After identifying this run, the run with the second highest overlap is aligned to it. This alignment involves a two-stage fitting process using a Generalized Additive Model (GAM) with splines. In the first stage, all overlapping peptides are used to fit a calibration curve that maps retention times

between the second and first run. Next, any overlapping peptides are removed that have a relative absolute error greater than 0.5% between retention times of the two runs after calibration. These remaining peptides are used to fit a more robust calibration curve. This second-stage calibration curve is then used to align all peptides (overlapping and non-overlapping) of the second run against the first. The median retention times of overlapping peptides are taken as the new representative retention time for that peptide sequence. This process is repeated iteratively until all runs are aligned, with the median taken regarding all previously aligned retention time values. After the alignment, peptide sequences are filtered on at least five observations and a maximum standard deviation of 2.0 iRT or minutes, ensuring only confident peptide and retention time pairs remain.

In order to evaluate the calibration procedure, a standard deviation is calculated for each peptidoforms across different LC-MS runs before calibration and after calibration. Each standard deviation is normalized by the elution length (retention time between the first and last observed peptide after calibration). This evaluation in Supplementary Fig. 1 shows that there is no substantial difference in standard deviation for the majority of data sets, but when calibration is required between runs they are properly aligned and show a mean or median standard deviation below 1%, lowering the standard deviation on average by 0.35% across all data sets.

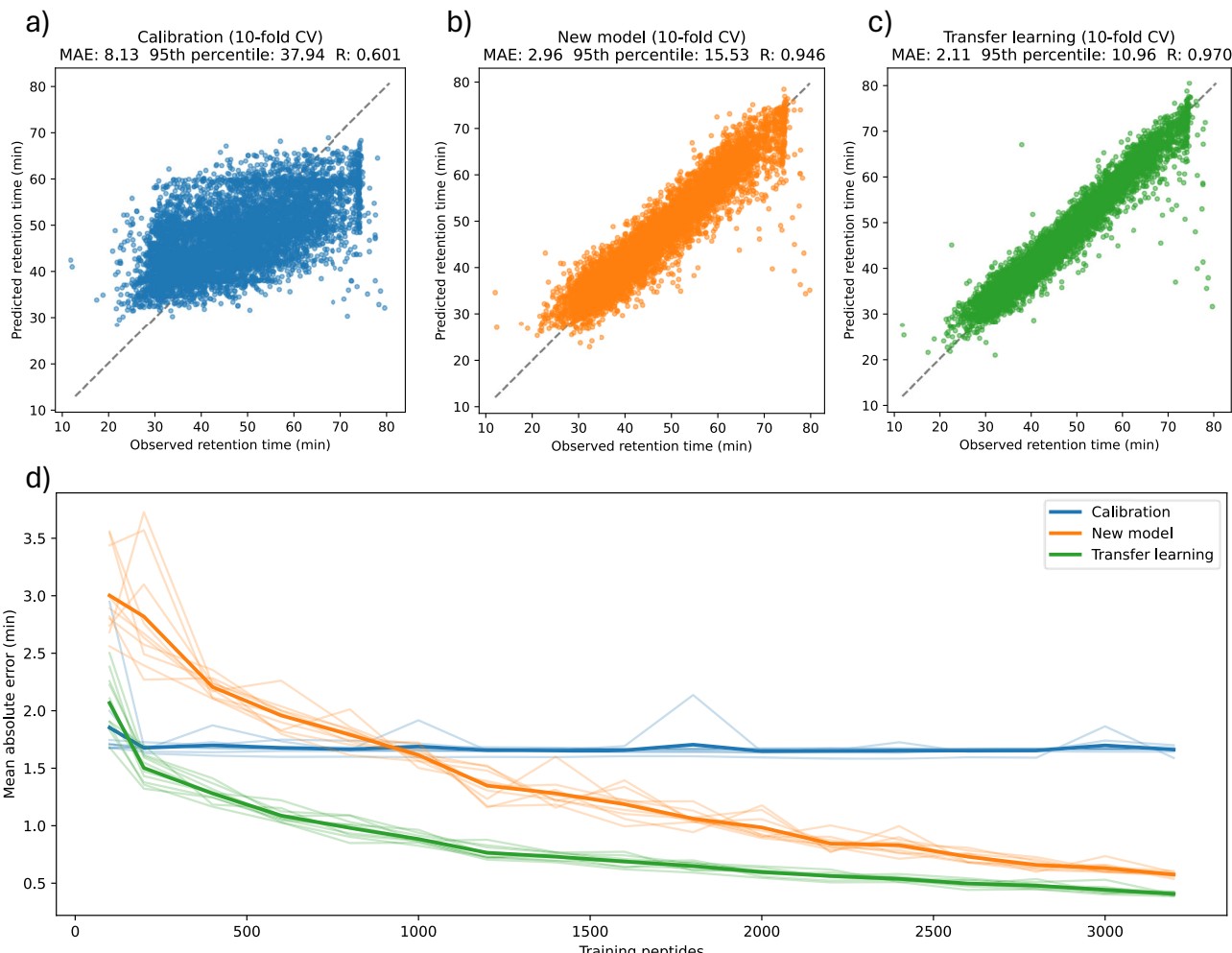

**Fig. 4 | Performance comparison for basic LC conditions.** Evaluation consists of **a** calibration, **b** new model (random parameter initialization), and **c** transfer learning. **d** Learning curve showing test set MAE against the number of training peptides for the basic LC conditions. Thin lines represent random seed replicas and thick lines represent the mean of these seed replicas.

Unless noted otherwise, each data set was randomly divided into a test set (10%), a validation set (5%), and a training set (85%). For each of the learning-curves, a single 75%/25% train-test split was used, then models were trained on training subsets that become gradually larger and tested on the same test set. This strategy for the learning curves was repeated for 10 different seeds. All models were trained for up to 300 epochs, with early stopping based on the validation set if available. Prediction performance was evaluated using three commonly used metrics: MAE, Pearson correlation (R), and Δt95%, which describes the error for a retention time window that contains 95% of the peptides in the error distribution. To ensure comparability between experiments, MAE and Δt95% were normalized by dividing them by the retention time difference between the first and last detected peptide in each data set. These metrics are referred to as relative MAE and relative Δt95%.

Each evaluation presented here compared three methods: Transfer learning, calibration with a GAM, and a model with randomly initialized parameters (a new model). In these evaluations the calibration method uses the training set to determine a mapping from predicted values from a pretrained model using a GAM with splines. In the case of training a new model from scratch, the model is initialized with random weights drawn from a normal distribution ($\mu = 0.0$, $\sigma = 1.0$). Finally, for transfer learning the parameters before training are set based on a model fitted on the data set from PXD005573. This data set was chosen as it is one of the largest data sets ($n = 113,213$) and most accurate models in DeepLC. It is important to note that the only difference between the models trained from scratch (with randomly initialized parameters) and those trained with transfer learning are the initial parameters. All model parameters remained trainable, as no layers were frozen, and hyperparameters remained the same.

### General training procedure
For each data set and prediction method three distinct models in DeepLC must be fitted, which differ in the kernel sizes for their convolutional layers, and these are then used in an ensemble that is averaged to obtain a single prediction per peptidoform. For transfer learning, the MD5 hash of the architecture is used to match the corresponding hyperparameters.

### Evaluation of a TMPP-labeling data set
To evaluate the performance of transfer learning in predicting retention times for peptides carrying substantially different modifications, an experiment with (N-succinimidyloxycarbonylmethyl)tris(2,4,6-trimethoxyphenyl)phosphonium bromide (TMPP) labeling was conducted. This very large, and rather atypical peptide modification (768.54 Da) causes a significant shift towards a more hydrophobic retention time, making it an effective test for transfer learning for previously unseen peptide modifications.

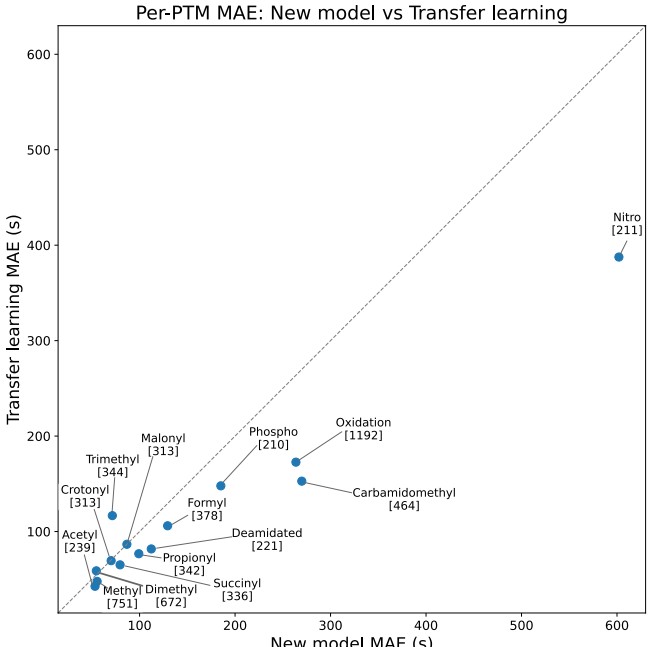

**Fig. 5 | Mean Absolute Error between a model with random starting parameters (new model) and one using transfer learning for fourteen PTMs from Zolg et al.[34].** The numbers beneath each PTM indicate the count of peptides carrying that modification in the test set.

The data from PXD064426 was used for this evaluation and a 10-fold cross validation (CV) approach was used due to the limited size of the data set ($n$ = 4505 peptidoforms) and to test the model robustness. In this study, the chromatographic gradient was optimized to improve elution of the more hydrophobic peptides with a 30 min gradient from 1 to 40% of organic phase (acetonitrile acidified with 0.1% formic acid). LC-MS/MS analyses were performed on a NanoAcquity LC-system (Waters, Milford, MA, USA) coupled to a Q-Exactive Plus Q-Orbitrap (Thermo Fisher Scientific, Waltham, MA, USA) mass spectrometer. The separation was performed using a Symmetry C18 precolumn (0.18 × 20 mm, 5 μm particle size, Waters and an ACQUITY UPLC® BEH130 C18 separation column (250 mm × 75 μm id, 1.7 μm particle size, Waters). Data was acquired using a Top10 Data Dependent Acquisition method and peptides were fragmented using higher energy collision dissociation fragmentation. Exact details and methodology of the study are described in Bertaccini et al.[31] and Vaca et al.[32].

### Evaluation of a highly basic LC setup
To evaluate the performance of transfer learning on a significantly different LC setup, an LC run under basic conditions (pH = 9) was used. These basic conditions drastically change the retention mechanism, providing an ideal testing ground for transfer learning's ability to adapt to new LC conditions. In this evaluation, similarly to the TMPP evaluation, a 10-fold CV approach was used to provide more reliable performance results due to the limited size of this data set ($n$ = 6363 peptidoforms).

HeLa S3 cells lysate was prepared and further digested by trypsin (Promega, Madison, WI, USA) and GluC (Roche, Mannheim, Germany) at 37 °C and 25 °C, respectively. Digested peptides were analyzed using an Orbitrap Fusion Lumos mass spectrometer (Thermo Scientific, San Jose, CA, USA) coupled to an UltiMate 3000 LC system (Thermo Fisher Scientific, Germering, Germany) operating in high-pH RP-HPLC mode (mobile phases: 2.5 mM imidazole and 3% IPA in water; 2.5 mM imidazole and 3% IPA in 95% acetonitrile). A self-packed analytical column

(XBridge Premier Peptide BEH C18, 2.5 μm, 75 μm i.d., 18 cm length, 130 Å; Waters, Milford, MA, USA) was used for the analysis. The mass spectrometer was operated in positive ion DDA mode; both MS1 and MS2 spectra were acquired in Orbitrap. The exact details and methodology of the study are described in Penanes et al.[33].

Mass spectrometric data were converted to mzML format by ThermoRawFileParser (1.3.4) and searched with MSGF+ (v2022.01.07) against Uniprot human proteome (20185 proteins) supplemented with reversed decoys. The results were rescored by Percolator (3.05). LC-MS elution profiles were produced by Biosaur2, and confident peptide-spectrum matches ($q$ < 0.01) were mapped to the features by accurate monoisotopic m/z (5 ppm tolerance), retention time, and charge. The apex retention time of the LC-MS feature was used as the elution time of the corresponding peptide. For the peptides detected more than once across the runs, the average elution time was used, and peptides with an elution time coefficient of variation above 10% were excluded. The data was deposited in PRIDE under accession number PXD064311 (Experiment 2, March).

### Evaluation of unseen modified peptides
DeepLC has the ability to predict retention times of unseen modifications accurately. This ability of DeepLC was previously evaluated with random starting parameters on fourteen modifications from Zolg et al.[34], this dataset contains a total of 5693 peptides. Here, the same analysis is performed, with the inclusion of transfer learning and calibration. Each modification is evaluated by excluding all peptides carrying that modification and using the remainder for training or calibration. The generalization performance of all three methods is then evaluated on the excluded peptides carrying the specific modification. For transfer learning and calibration, the model trained on PXD005573 is used.

### Reporting summary
Further information on research design is available in the Nature Portfolio Reporting Summary linked to this article.

### Data availability
The statistics for each LC-MS data is available in Supplementary Data 1. In addition to these data sets PXD064426 (TMPP) and PXD064311 (basic LC) are deposited in PRIDE. The PRIDE identifier for all prior analyzed datasets are PXD000044, PXD000239, PXD000258, PXD000511, PXD001820, PXD002549, PXD002929, PXD003093, PXD003698, PXD003879, PXD004186, PXD004269, PXD004356, PXD005271, PXD005346, PXD005499, PXD005509, PXD005514, PXD005664, PXD005751, PXD005766, PXD005849, PXD005850, PXD005857, PXD006115, PXD006522, PXD006638, PXD006644, PXD006702, PXD006717, PXD007168, PXD007546, PXD008005, PXD008039, PXD008048, PXD008250, PXD008289, PXD008476, PXD008495, PXD008574, PXD008599, PXD008726, PXD008798, PXD008863, PXD008916, PXD009262, PXD009424, PXD009464, PXD009621, PXD010098, PXD010099, PXD010133, PXD010248, PXD010249, PXD010576, PXD010606, PXD010709, PXD010773, PXD010819, PXD010895, PXD011143, PXD011252, PXD011518, PXD011545, PXD011863, PXD011921, PXD012121, PXD012147, PXD012332, PXD012544, PXD012584, PXD012650, PXD012681, PXD012891, PXD012975, PXD012997, PXD013056, PXD013129, PXD013307, PXD013340, PXD013485, PXD013507, PXD013590, PXD013738, PXD013923, PXD014119, PXD014247, PXD014324, PXD014325, PXD014525, PXD014552, PXD014561, PXD014619, PXD014691, PXD014794, PXD014893, PXD014998, PXD015352, PXD015442, PXD015453, PXD015736, PXD015993, PXD016003, PXD016004, PXD016054, PXD016190, PXD016383, PXD016438, PXD016565, PXD016632, PXD016675, PXD016745, PXD016750, PXD016786, PXD017056, PXD017478, PXD017523, PXD017541, PXD017579, PXD017597, PXD017821, PXD017932, PXD017959,

PXD018148, PXD018218, PXD018253, PXD018387, PXD018422, PXD018584, PXD018617, PXD018752, PXD018900, PXD018905, PXD019074, PXD019252, PXD019254, PXD019285, PXD019362, PXD019443, PXD019454, PXD019483, PXD019513, PXD019545, PXD019600, PXD019692, PXD019704, PXD019708, PXD019880, PXD019881, PXD019929, PXD019951, PXD019957, PXD019967, PXD020019, PXD020180, PXD020222, PXD020224, PXD020303, PXD020450, PXD020490, PXD020491, PXD020494, PXD020559, PXD020561, PXD020686, PXD020742, PXD020759, PXD020812, PXD020844, PXD020910, PXD020939, PXD020969, PXD020972, PXD020974, PXD020987, PXD020996, PXD021018, PXD021109, PXD021320, PXD021359, PXD021360, PXD021437, PXD021507, PXD021540, PXD021561, PXD021623, PXD021624, PXD021630, PXD021631, PXD021636, PXD021677, PXD021731, PXD021758, PXD021818, PXD021882, PXD021924, PXD022091, PXD022149, PXD022285, PXD022303, PXD022383, PXD022492, PXD022525, PXD022544, PXD022549, PXD022565, PXD022614, PXD022662, PXD022752, PXD022786, PXD022904, PXD022909, PXD022955, PXD022962, PXD022984, PXD022985, PXD023239, PXD023277, PXD023559, PXD023595, PXD023679, PXD023701, PXD023703, PXD023735, PXD023739, PXD023852, PXD023998, PXD024045, PXD024093, PXD024094, PXD024095, PXD024127, PXD024162, PXD024168, PXD024193, PXD024335, PXD024340, PXD024550, PXD024782, PXD024794, PXD024803, PXD024808, PXD024852, PXD024872, PXD024937, PXD025052, PXD025127, PXD025146, PXD025184, PXD025217, PXD025258, PXD025375, PXD025503, PXD025626, PXD025726, PXD025727, PXD025767, PXD025770, PXD025792, PXD025817, PXD025832, PXD025899, PXD025933, PXD025980, PXD026057, PXD026072, PXD026085, PXD026264, PXD026407, PXD026452, PXD026487, PXD026638, PXD026713, PXD026719, PXD026720, PXD026804, PXD026805, PXD026824, PXD026842, PXD026844, PXD026857, PXD026894, PXD027163, PXD027185, PXD027288, PXD027294, PXD027352, PXD027427, PXD027707, PXD027768, PXD027786, PXD027810, PXD028107, PXD028125, PXD028276, PXD028326, PXD028518, PXD028526, PXD028568, PXD028599, PXD028636, PXD028797, PXD028901, PXD028911, PXD028925, PXD028992, PXD029191, PXD029290, PXD029297, PXD029404, PXD029429, PXD029539, PXD029660, PXD029668, PXD029710, PXD029730, PXD029747, PXD029831, PXD029895, PXD030003, PXD030012, PXD030164, PXD030309, PXD030419, PXD030484, PXD030524, PXD030729, PXD030883, PXD030955, PXD031137, PXD031174, PXD031272, PXD031332, PXD031378, PXD031795, PXD031921, PXD031936, PXD031987, PXD032030, PXD032031, PXD032041, PXD032260, PXD032781, PXD032893, PXD032903, PXD032958, PXD032979, PXD033012, PXD033065, PXD033122, PXD033538, PXD033763, PXD034055, PXD034196, PXD034370, PXD034453, PXD034457, PXD034517, PXD035547, PXD035553, PXD035924, PXD036107. Source data, intermediate files, and trained models are provided in the following Zenodo: https://doi.org/10.5281/zenodo.15269422. Source data are provided with this paper.

## Code availability

DeepLC: https://github.com/compomics/DeepLC. DeepLCRetrainer: https://github.com/RobbinBouwmeester/DeepLCRetrainer/tree/main/deeplcretrainer. ProteomicsML data preparation: https://github.com/ProteomicsML/ProteomicsML/blob/1da6a13f4e9b2a6e249df98e41cd24f52607232b/tutorials/retentiontime/deeplc-transfer-learning.ipynb. Code to fully reproduce the analysis is available at: https://doi.org/10.5281/zenodo.15269422.

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

## Acknowledgements

R.D., A.D., R.G., L.M., and R.B. acknowledge funding from the Research Foundation Flanders (FWO) [1SH9O24N, 12AK526N, 1SE3724N, G010023N, G028821N, 12A6L24N, I002819N, W005325N]. A.N. acknowledges funding from the European Union's Horizon 2020 research and innovation program under the Marie Skłodowska-Curie grant agreement N° 956148. L.M. acknowledges funding from the Horizon Europe Projects BAXERNA 2.0 [101080544] and COMBINE [101191739], and from the Ghent University Concerted Research Action [BOF21/GOA/033]. L.M. and C.C. are further supported by the CHIST-ERA project ODEEP-EU [G0GDV23N]. TMPP experiments (M.R., C.C.) were supported by the French Proteomic Infrastructure (ProFI UAR2048, ANR-10-INBS08-03, ANR-24-INBS-0015). P.P. and F.K. acknowledge the support by the Lundbeck Foundation (Grant R346-2020-1215 to F.K.). Proteomics and mass spectrometry infrastructure at the University of Southern Denmark SDU was supported by generous grants to the VILLUM Center for Bioanalytical Sciences (VILLUM Foundation grant no. 7292), PRO-MS: Danish National Mass Spectrometry Platform for Functional Proteomics (grant no. 5072-00007B), and the Novo Nordisk Foundation (INTEGRA, NNF20OC0061575).

## Author contributions

R.B., A.N., and L.M. conceptualized the study. R.B., A.N., V.G., P.P., F.K., M.R., and C.C. performed the experiments. R.B., A.N., A.D., R.D., K.V., and R.G. analyzed the data and wrote the software. F.K., C.C., and L.M. provided resources, guided experimental design, supervised and funded the research. R.B. and A.N. wrote the original draft. R.B., A.N., A.D., R.D., K.V., V.G., P.P., F.K., M.R., C.C., R.G., and L.M. reviewed and edited the manuscript.

## Competing interests

The authors declare no competing interests.
