## [Transparent Peer Review file · Nature Communications]

Transfer learning in DeepLC improves LC retention time prediction across substantially different modifications and setups

Corresponding Author: Professor Lennart Martens

Version 0:

Reviewer comments:

Reviewer #1

(Remarks to the Author)

This study focuses on model fine-tuning (transfer learning) for retention time (RT) prediction in dataset- or run-specific analyses. The authors demonstrate that transfer learning achieves better prediction performance compared to conventional calibration or full re-training approaches. However, this concept has already been widely adopted in existing tools such as DeepRT, AutoRT, AlphaPeptDeep, and DiaNN. These tools are also able to fine-tune the model for PTMs. So the results did not particularly surprise me. While the authors evaluated their method on a large number of datasets, the methodology and findings do not present substantial novelty to the field.

Compared to other approaches, the distinguishing feature of DeepLC lies in its ability to predict RTs for peptides with unseen post-translational modifications (PTMs). As a natural continuation of this work, I believe DeepLC should further emphasize RT prediction for unseen PTMs following model fine-tuning based on "seen" PTMs. This would clearly set it apart from other methods. If this can be successfully demonstrated, the novelty and impact would be sufficient to merit publication in Nature Communications.

(Remarks on code availability)

I have been using DeepLC for several times, it worked well.

Reviewer #2

(Remarks to the Author)

Transfer learning is a machine learning technique that enables a deep learning model trained on large-scale data to adapt to a new data domain, allowing effective predictions even with limited new data. This paper builds upon the previously developed DeepLC model and applies transfer learning to predict peptide retention times in bottom-up proteomics data obtained from different liquid chromatography (LC) environments. The authors provide experimental results demonstrating that transfer learning yields superior performance compared to conventional calibration methods or training new models from scratch. Furthermore, by applying their approach to various public datasets, the authors show that the effectiveness of transfer learning is correlated with the size of the dataset used for fine-tuning.

1. (Lines 90–102) The manuscript describes an alignment method used to calibrate retention times across runs. However, it is unclear whether this method is a novel contribution of this paper or if it has been previously published and validated. If this method is based on prior work (e.g., from ProteomicsML), the relevant references should be cited. If it is newly proposed in this manuscript, additional explanation regarding its appropriateness and an evaluation of its performance would strengthen the paper.

2. The manuscript evaluates transfer learning using two specific datasets: one involving TMPP-labelled peptides and another obtained from a highly basic LC setup. However, the exact number of peptides in these datasets is not explicitly stated. While the approximate sizes can be inferred from Figures 3d and 4d, providing the exact dataset sizes in the main text would improve transparency and reproducibility.

3. (Line 198) The manuscript claims that transfer learning consistently shows a decrease in MAE as training data increases, compared to calibration as the size of the training data increases (Figure 1d–f). However, this trend appears to be evident primarily in Figure 1f, while Figures 1d and 1e show less consistent patterns. Additionally, the manuscript does not explain why PXD016632, PXD017478, and PXD002929 were selected from the 362 PRIDE projects for this comparison. Clarifying the rationale for choosing these datasets would help readers better understand the observed performance differences.

4. It is unclear whether the learning curves presented in Figures 1, 3, and 4 represent training error or test error. If the curves show training error, the corresponding performance on the test set should also be reported.

5. The manuscript states that all models were trained using a 10:5:85 split for test, validation, and training sets. However, cross-validation is a standard approach for robust performance evaluation, especially when datasets are small. For the relatively limited TMPP-labelled and basic LC datasets, k-fold cross-validation (e.g., 10-fold) would provide a more reliable assessment of model performance. The authors should consider including cross-validation results or justify why they chose not to use this approach.

(Remarks on code availability)

Reviewer #3

(Remarks to the Author)

(Remarks on code availability)

Version 1:

Reviewer comments:

Reviewer #1

(Remarks to the Author)

The issue of novelty remains inadequately addressed. I acknowledge that the revised manuscript now includes Figure 5 and supplementary figures on unseen post-translational modifications (PTMs). Nevertheless, the other figures primarily reanalyze existing data using the well-established methodology of transfer learning for retention time prediction.

While the study leverages an impressively large dataset from 362 PRIDE projects, a large-scale reanalysis is not inherently novel—particularly when the underlying approach echoes that of DeepRT (2018, <https://pubs.acs.org/doi/10.1021/acs.analchem.8b02386>). Crucially, this reanalysis does not appear to yield substantial new insights that go beyond the contributions of DeepRT and subsequent related work.

In their response, the authors highlight two scenarios for fine-tuning: (i) unfamiliar modifications (TMPP) and (ii) substantially different chromatographic setups. I am skeptical of both points. First, DeepRT already demonstrated the adaptation of models to different LC conditions via fine-tuning. Second, the handling of PTMs through transfer learning was also previously explored, as illustrated in AlphaPeptDeep's Figure 3b.

The authors further argued that their work is the first to show transfer learning in "so substantially different scenarios". This, however, describes a suitable application for transfer learning, not a novel scientific result. The technique is explicitly designed for such generalization; proving its necessity in challenging cases does not, by itself, demonstrate an advancement over the state of the art.

For this reason, I previously urged a focus on rigorously evaluating performance on genuinely "unseen" PTMs, where neither the specific modification nor chemical analogues were present in the training data of prior models. Unfortunately, the revision does not center on this critical direction, and it still lacks evidence that the work substantively progresses beyond the foundation laid by studies like DeepRT other AI-based RT prediction tools.

While the authors have made efforts to address the concerns, my overall assessment is that the advance described is incremental rather than transformative. The work is solid and would be of interest to the proteomics community, but its scope and level of conceptual novelty are better suited for a specialized journal. I therefore cannot recommend publication in Nature Communications.

(Remarks on code availability)

Reviewer #2

(Remarks to the Author)

1. In reviewing the fine-tuning results on unseen PTMs (author's response to reviewer 1), I have a question regarding the experimental setup. PTMs such as phosphorylation, oxidation, and carbamidomethylation are typically among the most abundant modification types across datasets. Were these PTMs entirely excluded from the training set and only used for testing in the unseen PTM evaluation? Additionally, could the authors provide the total number of instances for each PTM type in the dataset?

2. In the newly added Figure 5 in manuscript, in addition to the MAE scatter plot, it would be informative to assess whether the distributions between the transfer learning and the new model differ significantly across individual PTM types. It would also be helpful if the authors could include the corresponding p-values from statistical tests to support these comparisons.

3. A more detailed rationale for selecting the specific datasets among the 362 PRIDE projects would strengthen the manuscript. Notably, PXD002929 exhibits higher MAE despite an increased number of training peptides. Further analysis is recommended to explain whether this limited improvement is due to discrepancies between the pretrained dataset and PXD002929 in sample composition or experimental conditions.

(Remarks on code availability)

Reviewer #3

(Remarks to the Author)

(Remarks on code availability)

Reviewer #1

This study focuses on model fine-tuning (transfer learning) for retention time (RT) prediction in dataset- or run-specific analyses. The authors demonstrate that transfer learning achieves better prediction performance compared to conventional calibration or full re-training approaches. However, this concept has already been widely adopted in existing tools such as DeepRT, AutoRT, AlphaPeptDeep, and DiaNN. These tools are also able to fine-tune the model for PTMs. So the results did not particularly surprise me. While the authors evaluated their method on a large number of datasets, the methodology and findings do not present substantial novelty to the field.

We would like to thank the reviewer for their constructive and thoughtful review. The recommended analysis demonstrating possible unobserved modifications via transfer learning has strengthened the manuscript's conclusions.

We fully agree with the reviewer that transfer learning (or fine-tuning) has been previously shown to be beneficial for retention time prediction and is widely adopted. However, to the current day there has not been a study that comprehensively analyzed the impact of transfer learning on such a wide scale (i.e., 362 PRIDE projects). Furthermore, we show that transfer learning is not only beneficial for regular LC-MS proteomics workflows but extends to two highly relevant situations for fine-tuning: (i) modifications (TMPP) that are very different from anything trained on and (ii) substantially different chromatographic setups (basic conditions). Importantly, there is currently no paper that shows the ability of models to transfer learn to these scenarios that are so substantially different.

Compared to other approaches, the distinguishing feature of DeepLC lies in its ability to predict RTs for peptides with unseen post-translational modifications (PTMs). As a natural continuation of this work, I believe DeepLC should further emphasize RT prediction for unseen PTMs following model fine-tuning based on “seen” PTMs. This would clearly set it apart from other methods. If this can be successfully demonstrated, the novelty and impact would be sufficient to merit publication in Nature Communications.

We fully agree with the reviewer that adding the effect of transfer learning for (unobserved) modifications would strengthen the manuscript. Therefore, we performed similar evaluations as previously presented in DeepLC¹. Specifically, the performance for unseen modifications with and without fine-tuning is compared on a dataset with synthetic modified peptides². For training, one of the modifications was removed from the data and is used as the test set. For each modification a previously trained model (“Calibration”), a model with randomly initialized parameters (“New model”), and a model fine-tuned (“Transfer Learning”) on the synthetic peptide data are then compared.

The results show that there is a modest, but overall, mostly consistent improvement in performance for “Transfer learning” (Figure 1 in this document; Figure 5 in the manuscript). The only exception (out of the fourteen tested) is trimethyl where the model trained from scratch shows a much lower mean absolute error (MAE). The figure along with its text has been added to the results section of the manuscript. As “Calibration” performed substantially worse, we opted to include “Calibration” only in the supplemental figure (Figure 2 in this document; Supplementary Figure 19 in the manuscript).

We would like to thank the reviewer for the insightful comment. We feel that the new analysis has substantially strengthened the conclusions of the manuscript.

Figure 1: Mean Absolute Error between a model with random starting parameters (new model) or transfer learning for fourteen PTMs from Zolg et al.².

Figure 2: Error for models with random starting parameters (new model), calibration (pretrained model that is calibrated) or transfer learning for fourteen PTMs from Zolg et al.².

Reviewer #1 (Remarks on code availability):

I have been using DeepLC for several times, it worked well.

Reviewer #2

Transfer learning is a machine learning technique that enables a deep learning model trained on large-scale data to adapt to a new data domain, allowing effective predictions even with limited new data. This paper builds upon the previously developed DeepLC model and applies transfer learning to predict peptide retention times in bottom-up proteomics data obtained from different liquid chromatography (LC) environments. The authors provide experimental results demonstrating that transfer learning yields superior performance compared to conventional calibration methods or training new models from scratch. Furthermore, by applying their approach to various public datasets, the authors show that the effectiveness of transfer learning is correlated with the size of the dataset used for fine-tuning.

We would like to thank the reviewer for their constructive and thoughtful comments. The additional analysis regarding the calibration and cross-validation have strengthened the conclusions of the manuscript.

1. (Lines 90–102) The manuscript describes an alignment method used to calibrate retention times across runs. However, it is unclear whether this method is a novel contribution of this paper or if it has been previously published and validated. If this method is based on prior work (e.g., from ProteomicsML), the relevant references should be cited. If it is newly proposed in this manuscript, additional explanation regarding its appropriateness and an evaluation of its performance would strengthen the paper.

This method is described in ProteomicsML however, it was not evaluated on a large amount of data. Therefore, we agree with the reviewer that this evaluation should still be performed. Furthermore, the amount of data tested in this manuscript allowed us to do a more thorough evaluation of the calibration approach.

For this analysis we are comparing the standard deviation across runs for the same peptidofoms (Figure 3 in this document; Supplementary Figure 1 in the manuscript). For each peptidofom we obtain a standard deviation across different LC-MS runs before calibration and after calibration. Each standard deviation is normalized by the elution length (retention time between the first and last observed peptide after calibration). To summarize the normalized standard deviations, we took either the mean or median value per dataset.

To our own surprise most of the mean or median standard deviations did not change before or after calibration. This can be explained by MaxQuant already aligning the different runs. However, if runs are not grouped by the user in MaxQuant, calibration is still required. The evaluation of the calibration shows that there is no difference for the majority of datasets, but when calibration is needed they are properly aligned and show a mean or median standard deviation below 1%.

Figure 3: Median (a and b) and mean (c and d) standard deviation between peptidiforms belonging to the same dataset, before and after the calibration procedure between LC-MS runs. Where either a direct comparison between datasets (a and c) is made or the delta in standard deviation is shown (b and d).

2. The manuscript evaluates transfer learning using two specific datasets: one involving TMPP-labelled peptides and another obtained from a highly basic LC setup. However, the exact number of peptides in these datasets is not explicitly stated. While the approximate sizes can be inferred from Figures 3d and 4d, providing the exact dataset sizes in the main text would improve transparency and reproducibility.

Indeed, this would improve transparency and reproducibility. We also feel that the other Figures could benefit from this information, so we included a table that included all dataset sizes and metrics as a supplementary table 1.

3. (Line 198) The manuscript claims that transfer learning consistently shows a decrease in MAE as training data increases, compared to calibration as the size of the training data increases (Figure 1d-f). However, this trend appears to be evident primarily in Figure 1f, while Figures 1d and 1e show less consistent patterns. Additionally, the manuscript does not explain why PXD016632, PXD017478, and PXD002929 were selected from the 362 PRIDE projects for this comparison. Clarifying the rationale for choosing these datasets would help readers better understand the observed performance differences.

These datasets were randomly chosen after first sorting on their respective number of datapoints. The goal was to have a subset that contained many peptides while still showing some variety in terms of datapoints.

4. It is unclear whether the learning curves presented in Figures 1, 3, and 4 represent training error or test error. If the curves show training error, the corresponding performance on the test set should also be reported.

The show performance is the test error, we have clarified the text at in the methods and the figure captions.

5. The manuscript states that all models were trained using a 10:5:85 split for test, validation, and training sets. However, cross-validation is a standard approach for robust performance evaluation, especially when datasets are small. For the relatively limited TMPP-labelled and basic LC datasets, k-fold cross-validation (e.g., 10-fold) would provide a more reliable assessment of model performance. The authors should consider including cross-validation results or justify why they chose not to use this approach.

We agree with the reviewer that showing different splits for training and testing with cross-validation (CV) can help to get a more robust performance estimate for small datasets. We have re-run all analysis for TMPP and basic LC with a 10-fold CV (Figures 4 and 5 in this document; Figures 3 and 4 in the manuscript). The conclusions remained the same, but we are now much more confident in concluding that transfer learning outperforms training a model from scratch or calibration. Furthermore, TMPP shows folds with a visible deviation from the expected trend for the learning curve. Indeed, this shows dependence on the exact selected subset and importance of CV.

Figure 4: Evaluation of predicted performance on TMPP labeled peptides.

Figure 5: Evaluation of predicted performance on basic LC conditions.

Reviewer #3 (Remarks to the Author):

References

1. Bouwmeester, R., Gabriels, R., Hulstaert, N., Martens, L. & Degroev, S. DeepLC can predict retention times for peptides that carry as-yet unseen modifications. *Nature Methods* 2021 18:11 **18**, 1363–1369 (2021).
2. Paul Zolg, D. *et al.* Proteometools: Systematic characterization of 21 post-translational protein modifications by liquid chromatography tandem mass spectrometry (lc-ms/ms) using synthetic peptides. *Molecular and Cellular Proteomics* **17**, 1850–1863 (2018).

REVIEWER COMMENTS

Reviewer #1 (Remarks to the Author):

The issue of novelty remains inadequately addressed. I acknowledge that the revised manuscript now includes Figure 5 and supplementary figures on unseen post-translational modifications (PTMs). Nevertheless, the other figures primarily reanalyze existing data using the well-established methodology of transfer learning for retention time prediction.

While the authors have made efforts to address the concerns, my overall assessment is that the advance described is incremental rather than transformative. The work is solid and would be of interest to the proteomics community, but its scope and level of conceptual novelty are better suited for a specialized journal. I therefore cannot recommend publication in Nature Communications.

We thank the reviewer for acknowledging that the work is solid. However, we wholeheartedly disagree with the reviewer's statement that the work is merely incremental. Below, we provide a point-by-point reply to the reviewer's concerns.

While the study leverages an impressively large dataset from 362 PRIDE projects, a large-scale reanalysis is not inherently novel—particularly when the underlying approach echoes that of DeepRT (2018, <https://pubs.acs.org/doi/10.1021/acs.analchem.8b02386>). Crucially, this reanalysis does not appear to yield substantial new insights that go beyond the contributions of DeepRT and subsequent related work.

While the DeepRT paper presents a total of eight datasets—a number that is considerably smaller, and likely far less diverse, than the 362 included in our study—it is also important to note that only two datasets were evaluated in the DeepRT paper for transfer learning (the ones named “Hela” and “yeast”). We believe that this limited evaluation does not provide a sufficient basis for applying transfer learning in future research, given the substantial variability in LC conditions across datasets. Instead, the breadth of datasets in our manuscript does in fact enable conclusions about transfer learning to be generalized, thanks to the large amount of data. Moreover, and importantly, we also consider three distinct scenarios: (i) unseen modifications, (ii) substantially different LC conditions (acidic vs basic), and (iii) substantially different modifications (TMPP). Note that this covers essentially all possibilities: (i) variation in biology, (ii) very strong variation in experimental protocol, and (iii) highly complex modifications. We thus show that DeepLC can adapt to biology, can adapt to changes in workflow and protocol, and can adapt extremely flexibly to very large and complex peptide modifications. In summary, although DeepRT did apply transfer learning, its conclusions remain very restricted, which is understandable, as this was not the main point of the DeepRT paper. Rather, the emphasis was there on illustrating the possibility. Our manuscript, in contrast, establishes a much better foundation for the actual applicability and versatility that future research can refer to, and also adds a showcase for the specific ability of DeepLC to adapt to arbitrary modifications.

As such, we firmly believe that there is a considerable advance here, and that our manuscript establishes an important baseline for transfer learning capability on a very wide data set that

covers all relevant aspects. Indeed, we are convinced that our manuscript, and its extensive and varied datasets, will become the gold standard to which all other future efforts will be benchmarked, something that is unlikely to happen for the DeepRT paper and dataset.

In their response, the authors highlight two scenarios for fine-tuning: (i) unfamiliar modifications (TMPP) and (ii) substantially different chromatographic setups. I am skeptical of both points. First, DeepRT already demonstrated the adaptation of models to different LC conditions via fine-tuning. Second, the handling of PTMs through transfer learning was also previously explored, as illustrated in AlphaPeptDeep's Figure 3b.

While DeepRT applied transfer learning to two datasets, the base model and the data transfer learned to, concern RP datasets run at the same pH, using the same solvents (A and B), and identical column chemistry. In fact, this is essentially trivial in comparison to the wide variety of conditions in our manuscript. As explained above, we explore experiments where the column chemistry, solvents, and pH differ very substantially between experiments, actually taxing the transfer learning capability of the algorithm. Notably, the drastic change in pH has a significant impact on all peptide retention times (<https://doi.org/10.1016/j.chroma.2005.03.121>). Therefore, the transfer learning performed in DeepRT on two datasets is simply not comparable to the transfer learning applied in our 'basic pH' LC setup experiment, which demonstrates a distinctly different and quite important aspect of transfer learning. As highlighted in our manuscript, it is noteworthy that even with a complete change in the retention mechanism, transfer learning still manages to substantially improve performance. This has never been shown in any previous manuscript, and is highly interesting to experimentalists. Indeed, the message to proteomics researchers is very clear: thanks to transfer learning in DeepLC, you can now drastically alter the experimental conditions, and still be able to rely on accurate and precise retention time prediction. The importance of such experimental flexibility should not be underestimated, as the proteome is difficult to analyse precisely because of the enormous physico-chemical diversity of proteins, necessitating different protocols to analyse the full spectrum of proteins.

The reviewer also points out that modification analysis has already been demonstrated for AlphaPeptDeep. This is correct; however, for AlphaPeptDeep, this was performed on only a very limited set of relatively standard modifications; in fact, the same as we show in our manuscript's Figure 5. The TMPP evaluation is fundamentally different from this prior analysis, as TMPP is far more complex, and causes a far greater shift in retention time, compared to the modifications in the AlphaPeptDeep evaluation. Indeed, most modifications depicted in Figure 5 of our manuscript (or Figure 3 in the AlphaPeptDeep paper) cause only a fairly small retention time shift. As above, therefore, our evaluation goes very far beyond any previous work by truly challenging the transfer learning with TMPP, which, unlike the previously used modifications in the AlphaPeptDeep paper, dramatically alters the physicochemical properties of the peptides. This is clear from its large size (the molecular weight of TMPP is 572.18 Da, larger than five times an average amino acid), and from its many different functional hydrophobic groups such as aromatic phenyl rings and methoxy groups on the one hand, and hydrophilic groups such as carbonyl and phosphonium on the other hand. In short, such a molecule is truly outlandish to a retention time prediction algorithm, and it is a clear testament to the impressive capabilities of DeepLC that it can adapt to this molecule with such ease using transfer learning. Therefore, just as with the different experimental setups discussed above, the AlphaPeptDeep analyses are simply not comparable to ours and this distinction is moreover clearly emphasized in our manuscript.

The authors further argued that their work is the first to show transfer learning in "so substantially different scenarios". This, however, describes a suitable application for

transfer learning, not a novel scientific result. The technique is explicitly designed for such generalization; proving its necessity in challenging cases does not, by itself, demonstrate an advancement over the state of the art.

The distinction between what constitutes ‘a novel scientific result’ is often unclear, and may well be in the eye of the beholder. We respectfully disagree with the reviewer that novelty in research is exclusively achieved through the introduction of entirely new methods. Indeed, this definition would disqualify many highly cited papers that were extremely impactful to the field of proteomics. Here, the application of one of the most used retention time prediction algorithms for peptides in completely novel and highly challenging scenarios establishes an important foundation for the actual use of transfer learning as a means to drastically extend the applicability of a retention time algorithm through transfer learning. We are confident that our manuscript will serve as the reference work for the application of transfer learning to LC and future ML applications in proteomics, as we set a novel benchmark for this technology that goes very far beyond the limited proof-of-concept efforts the reviewer highlights. It is our strong opinion that comprehensive analysis and validation of methods in novel and thoroughly challenging conditions that really puts methods to the test, represent far more genuine scientific progress than mere incremental improvement where an existing machine learning technique (transfer learning did not originate with the DeepRT nor the AlphaPeptDeep papers) is applied in a non-challenging context to provide basic proof-of-concept.

For this reason, I previously urged a focus on rigorously evaluating performance on genuinely "unseen" PTMs, where neither the specific modification nor chemical analogues were present in the training data of prior models. Unfortunately, the revision does not center on this critical direction, and it still lacks evidence that the work substantively progresses beyond the foundation laid by studies like DeepRT other AI-based RT prediction tools.

There appears to be a misunderstanding regarding our manuscript’s intent. Our aim is to evaluate the performance and applicability of transfer learning across a wide range of scenarios. While including other modifications could be of biological interest, our focus is on using these modifications—such as TMPP—as representative cases to evaluate transfer learning. In fact, we explain in detail above that TMPP is essentially exactly what the reviewer demands: a genuinely unseen PTM, of a complexity, chemical character, and size that was never present in any of the other datasets or models. Indeed, as noted above, the research presented here is distinct from previous evaluations in the literature, providing an important, and much-needed foundation for future applications of transfer learning.

Furthermore, we would like to emphasise that the analysis presented in Figure 5 concerns modifications that were not part of the training data of the algorithm. Whenever we show transfer learning outcomes, these are tested on one or more modifications that the model being tested has never seen before. This is a very basic aspect of AI-model evaluation in general, of course, but, in light of the comment by the reviewer, we felt it important to point this out here. Thus, our response above assumes that the reviewer is up-to-date with this basic testing procedure, and we therefore interpret their comment as meaning that the assessment should focus on modifications not previously used for training of any other model or paper. In fact, this is true for TMPP, as this is the first instance where a model has been trained on this distinctly different modification, providing a true outgroup example.

While the authors have made efforts to address the concerns, my overall assessment is that the advance described is incremental rather than transformative. The work is solid and would be of interest to the proteomics community, but its scope and level of conceptual novelty are better suited for a specialized journal. I therefore cannot recommend publication in Nature Communications.

As we have documented in detail above, our manuscript is the first to truly test the capabilities of transfer learning for peptide retention time modelling under seriously challenging conditions (as opposed to simple proof-of-principle results shown previously), and this in all relevant possible ways (biological diversity, protocol diversity, modification diversity). Our manuscript therefore sets the first serious benchmark for transfer learning in predictive modelling in proteomics, and moreover sets a quite high performance bar to clear for future such models, which will undoubtedly render it highly impactful for the field at large. Incidentally, the other reviewers seem to agree with this view. To deny our manuscript the exposure it will get in Nature Communications therefore seems counterproductive, and not well substantiated.

Reviewer #2 (Remarks to the Author):

1. In reviewing the fine-tuning results on unseen PTMs (author's response to reviewer 1), I have a question regarding the experimental setup. PTMs such as phosphorylation, oxidation, and carbamidomethylation are typically among the most abundant modification types across datasets. Were these PTMs entirely excluded from the training set and only used for testing in the unseen PTM evaluation? Additionally, could the authors provide the total number of instances for each PTM type in the dataset?

First, we would like to thank the reviewers with their thoughtful remarks and questions. In response to their remarks, we confirm that even the most abundant PTMs were entirely excluded from the training set and only used for testing evaluation, we clarified this part better in the relative method section.

Additionally, we have included the number of peptides carrying each PTM beneath the respective PTM in figure 5. We also mention the total number of peptides in the dataset (5693 peptides), which provides the reader with the information on training and test set sizes for each PTM.

2. In the newly added Figure 5 in manuscript, in addition to the MAE scatter plot, it would be informative to assess whether the distributions between the transfer learning and the new model differ significantly across individual PTM types. It would also be helpful if the authors could include the corresponding p-values from statistical tests to support these comparisons.

We have previously shown the distribution between transfer learning, training a model from scratch, and calibration in the supplementary figure 19. In the updated figure 19, we have also added the p-values for each PTM beneath the corresponding PTM to include statistical comparison.

3. A more detailed rationale for selecting the specific datasets among the 362 PRIDE projects would strengthen the manuscript. Notably, PXD002929 exhibits higher MAE despite an increased number of training peptides. Further analysis is recommended to explain whether this limited improvement is due to discrepancies between the pretrained dataset and PXD002929 in sample composition or experimental conditions.

Datasets were selected based on the availability of MaxQuant result files. This selection was applied as there is a large number of datasets that have these identification files available. Furthermore, MaxQuant determines the elution apex, which is important for accurate retention time prediction. We have described this selection procedure in the methods section of the manuscript in more detail.

We have inspected PXD002929 and it seems there is a gap in the chromatogram where no peptides are identified (or quantified). This could be due to issues with either the LC or MS, and thus this results in lower performance in the region between 100 and 150 retention time. Importantly, the conclusion for this dataset is the same as all the other ones; transfer learning improves over calibration and starting with random parameters.